# FROM FRAMES TO CLIPS: EFFICIENT KEY CLIP SELECTION FOR LONG-FORM VIDEO UNDERSTANDING

## ABSTRACT

Video Large Language Models (VLMs) have achieved remarkable results on a variety of vision language tasks, yet their practical use is limited by the "needle in a haystack" problem: the massive number of visual tokens produced from raw video frames exhausts the model's context window. Existing solutions alleviate this issue by selecting a sparse set of frames, thereby reducing token count, but such frame-wise selection discards essential temporal dynamics, leading to suboptimal reasoning about motion and event continuity. In this work we systematically explore the impact of temporal information and demonstrate that extending selection from isolated key frames to key clips - which are short, temporally coherent segments - improves video understanding. To maintain a fixed computational budget while accommodating the larger token footprint of clips, we propose an adaptive resolution strategy that dynamically balances spatial resolution and clip length, ensuring a constant token count per video. Experiments on three long-form video benchmarks demonstrate that our training-free approach, F2C, outperforms uniform sampling up to $8.1\%, 5.6\%$, and $10.3\%$ on Video-MME, LongVideoBench and MLVU benchmarks, respectively. These results highlight the importance of preserving temporal coherence in frame selection and provide a practical pathway for scaling VLMs to real world video understanding applications.

## 1 INTRODUCTION

Video Large Language Models (VLMs) have demonstrated strong capabilities in video understanding tasks such as video question answering (VQA) and captioning (Maaz et al., 2023; Lin et al., 2023b; Zhang et al., 2023; Lin et al., 2023a; Jin et al., 2024; Ren et al., 2024; Liu et al., 2024b). By integrating visual encoders with Large Language Models (LLMs), VLMs can reason over multimodal content and achieve impressive performance (Yang et al., 2023; Chen et al., 2024; Wu et al., 2024a; Kim et al., 2024; Min et al., 2024). However, processing long-form videos remains a major bottleneck due to the overwhelming number of visual tokens relative to language tokens. For instance, a video with 1K resolution (1024×768) produces 4,015 tokens per two frames when processed by Qwen2.5-VL (Bai et al., 2025). Even when sampled at only 1 FPS, a one-hour video yields over 7 million tokens, far exceeding the context length limits of current LLMs. Beyond computational constraints (Yu et al., 2024; Shen et al., 2024), excessive visual input can distract model attention, making it difficult to identify relevant content when facing more than 2k tokens (Cheng et al., 2025). This challenge is commonly referred to as the *Needle in a Haystack* problem.

To mitigate this, most VLMs adopt *uniform sampling* to reduce the number of frames before encoding. While effective for short video clips, uniform sampling assumes equal importance across all frames, which is rarely the case in long and uncurated videos such as surveillance or instructional recordings. In such scenarios, uniformly sampled frames often capture irrelevant content, degrading downstream performance (Li et al., 2023; Shu et al., 2024; Liu et al., 2025).

Recent works have explored keyframe selection as a form of *context management* for long videos (Ataallah et al., 2024; Yu et al., 2024; Shen et al., 2024). These methods typically rank frames based on visual similarity to the query and select the most relevant ones. Although this improves efficiency, the approaches overlook temporal continuity. In long videos, sparse keyframe sampling can leave large temporal gaps. For instance, 32 frames sampled from a 30-minute video correspond to roughly 50 seconds between frames, which loses motion cues and event progression,

*What direction(s) does the Ping Pong ball rotate in?*

*A. Clockwise throughout.*
*B. No rotation.*
*C. Clockwise then counter-clockwise.*
*D. Counter-clockwise throughout.*
*E. Counter-clockwise then clockwise.*

**(a) Single Key Frame**

**(b) Key Frame → Key Clip**

*Answer: D*

Figure 1: **Motivation of key clip selection.** In long-form videos (*e.g.*, 30 minutes), uniform sampling into 32 frames yields gaps of more than 50 seconds, so only a single frame may be selected from an event. As an example from Shangguan et al. (2025) shown, relying on one key frame (a) makes it impossible to answer motion-related questions such as the rotation direction of a Ping Pong ball. By extending a key frame into a short key clip (b), temporal continuity is preserved, enabling the correct reasoning.

as shown in Figure 1. This limitation raises a natural question: *Can temporal information from frames adjacent to the key frames improve downstream performance in VLMs?*

In this work, we revisit the frame selection problem from a new perspective. Instead of relying solely on isolated frames, we propose selecting *key clips*, which are short temporal segments centered around key frames. This preserves both semantic relevance and local temporal continuity. However, naively adding more frames increases the number of visual tokens, intensifying the memory bottleneck. A practical solution to this problem is to compensate frames with *lower spatial resolutions* for longer clip length. This introduces resolution as an additional factor in frame selection. By lowering resolution, one can incorporate longer clips without increasing the total token count. Thus, a principled trade-off emerges between clip length and spatial resolution, allowing a spatio-temporal balance.

Motivated by these insights, we propose **Frames-to-Clips** (F2C), a training-free framework for key clip selection with adaptive resolution to avoid additional computation. F2C enhances temporal coherence while maintaining computational efficiency, offering a practical solution to scaling VLMs for long-form video understanding. Through empirical analysis on long-form VQA benchmarks, we compare F2C with other frame selection methods and find that the temporal continuity in key clips consistently improves performance, underscoring the importance of temporal context in long-form video understanding.

Our contributions are three-fold:

- We conduct a comprehensive analysis of frame selection strategies and show that *selecting temporally coherent clips*, rather than isolated frames, significantly improves VLM performance on long-form videos.
- We propose a key clip selection framework, F2C, that integrates adjacent frames to restore temporal continuity, together with an adaptive resolution scheme to mitigate token overhead.
- *Without any additional training*, F2C outperforms uniform sampling up to 8.1%, 5.6%, and 10.3% on Video-MME (Fu et al., 2024), LongVideoBench (Wu et al., 2024b), and MLVU (Zhou et al., 2024), providing an effective and scalable solution for long-form video understanding.

## 2 RELATED WORK

### 2.1 VIDEO LARGE LANGUAGE MODEL FOR LONG-FORM VIDEO

Recent advances in Video Large Language Models (VLMs) have extended the multimodal reasoning ability of LLMs to video understanding tasks such as question answering and captioning. Many works improve the backbone architecture to better handle long-form inputs: Video-LLaVA (Lin et al., 2023a), Qwen2.5-VL (Bai et al., 2025), and VideoChat2 (Li et al., 2023) adapt image-language models with temporal modules or tailored training strategies; Chat-UniVi (Jin et al., 2024) and Video-oLLAMA2 (Cheng et al., 2024) refine multimodal fusion and instruction tuning; LLaVA-NeXT-QW2 (Liu et al., 2024a) and LLaVA-OneVision (Li et al., 2024) enhance vision encoders or adopt multi-granularity features; and LongVILA (Xue et al., 2024), LongVA (Zhang et al., 2024c), Video-XL (Shu et al., 2024), Video-CCAM (Fei et al., 2024), and LongVU (Shen et al., 2024) explicitly

target long videos through hierarchical attention, temporal compression, or memory mechanisms. Despite these advances, most approaches still rely on *uniform sampling* to reduce raw videos into a fixed number of frames, overlooking redundancy and the uneven distribution of relevant information. In contrast, we focus on this preprocessing stage and propose a context management strategy that selects temporally coherent clips with adaptive resolution, offering more informative and efficient inputs without modifying the model architecture.

## 2.2 TRAINING-BASED FRAME SELECTION

Learning-based approaches have been proposed to select the most informative frames for VLMs, though they are typically limited to short videos due to computational cost. ATP (Buch et al., 2022) and FFS (Buch et al., 2025) train selectors end-to-end using downstream task losses, while Frame-Voyager (Yu et al., 2024) ranks frame candidates by their task loss. Hu et al. (2025) leverage a strong vision-language model to generate pseudo-labels for supervision. ViaRL (Xu et al., 2025), ReFo-CUS (Lee et al., 2025) and SeViLA (Yu et al., 2023) apply reinforcement learning and self-learning to improve grounding before answering. GenS (Yao et al., 2025) and Chain-of-Frames (Ghazanfari et al., 2025) introduce datasets with keyframe annotations to enable supervised training. Although effective in controlled settings, these methods are resource-demanding to train and scale poorly to long-form videos, making training-free selection strategies more practical.

## 2.3 TRAINING-FREE FRAME SELECTION

Several methods have explored training-free frame selection. BOLT (Liu et al., 2025) is an early approach that leverages inverse transform sampling to improve diversity in selected frames. TCoT (Arnab et al., 2025) later utilized the long context window of Gemini-1.5-Flash (Team et al., 2024) to reason over candidate keyframes before answering. MDP3 (Sun et al., 2025) advanced this direction with a list-wise selection strategy that balances query relevance, diversity, and sequentiality. AKS (Tang et al., 2025) formulates keyframe selection as an optimization over prompt relevance and video coverage, and proposes an adaptive algorithm that selects informative keyframes to maximize retained information under fixed token budgets. Most recently, Q-Frame (Zhang et al., 2025), which is the most related to our work, incorporated dynamic resolution by ranking frames into multiple resolution levels. In contrast, we rethink frame selection from a dynamic perspective and propose an adaptive strategy over key clip length rather than treating frames as isolated units.

## 3 PRELIMINARY

### 3.1 PROBLEM FORMULATION

In this work, we consider a Video Question Answering (VQA) task for a video $V$ represented with $N$ frames $V = \{f_1, f_2, \ldots, f_N\}$ and an associated text query $Q$. VQA aims to generate an answer $A$ that accurately addresses the query based on the content in the given video. Our goal is that given $(V, Q)$, we want to sample critical frames to generate the answer with the accurate information.

Due to the limitation of the computation and memory, the VLM solutions for VQA usually down-sample the video into $K$ frames based on the hardware constraints. $K$ will usually be 8 to 32 frames, *i.e.*, $K \ll N$. By default, the frames will be sampled *uniformly*. Selected frames are then passed to a VLM along with the text query $Q$ to get an answer. Formally, a VLM represented by $f_{\text{VLM}}$ generates an answer for the given input as

$$A = f_{\text{VLM}}(Select(V), Q). \tag{1}$$

In this work, our goal is to perform input selection, *Select*, to provide more useful information to the VLM. Specifically, our task is formulated as choosing a set of selected frames $\{f'_1, f'_2, \ldots, f'_K\}$, where $f'_i \in \mathbb{R}^{H \times W}$, together with their spatial resolution represented by $H$ and $W$.

So far, we only select frame with its associated spatial resolution. However, temporal information can be crucial to answer certain questions (see Figure 1). We can incorporate temporal information, *key clips*, into our input for VLM as $\{C_1, C_2, \ldots, C_K\}$, where $C_i \in \mathbb{R}^{H \times W \times L}$. In this study, we propose an adaptive approach to incorporate *key clips* with adaptive spatial $(H, W)$ and temporal resolution $(L)$.

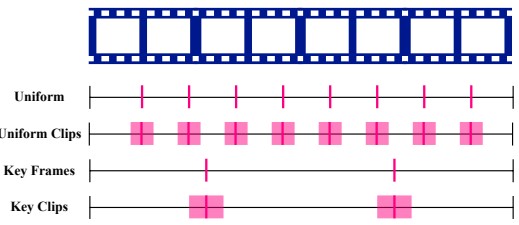

**Figure 2:** Uniform Sampling samples frames evenly from the video, and Key Frames are human-annotated crucial frames to answer the question. Uniform and Key Clips augment frames with their neighboring frames.

Table 1: Comparison of different sampling methods on the Ego4D-Haystack dataset using Qwen2.5-VL-7B.

| Method | # Frames | QA Accuracy % |
|---|---|---|
| Uniform | 32 | 45.2 |
| Uniform | 64 | 49.8 |
| Uniform Clips | $32 \times 2$ | 50.4 |
| Key Frames | 3-4 | 63.0 |
| Key Clips | 32 | 65.1 |

### 3.2 Importance of Key Frames and Temporal Information

To investigate the impact of key frames and the temporal information in VQA, we conduct an empirical study on the Ego4D-Haystack (Ye et al., 2025) dataset, which provides *human-annotated ground-truth* key frames for each video containing sufficient information to answer the question. This dataset contains long-form videos, each approximately 30 minutes in duration, which requires demanding capability on the long-context understanding. We use Qwen2.5-VL-7B as the downstream VLM to evaluate performance on the VQA task under uniform sampling, ground-truth key frames, and augmented key clips, as shown in Figure 2.

As shown in Table 1, selecting key frames based on human annotations yields a substantial improvement over uniform frame sampling, underscoring the importance of constructing task-relevant visual context. Note that even with only 3-4 key frames, key frames selection can outperform uniform sampling even with 64 frames, highlighting the strong effectiveness of selecting related frames.

To further investigate the role of temporal information, we extend both uniform sampling and keyframe selection by including neighboring frames to form short, temporally coherent segments, denoted as *uniform clips* and *key clips*. For uniform sampling, each frame is augmented into a 2-frame clip, while for keyframe selection we expand to a total of 32 frames for fair comparison. This simple augmentation consistently improves VQA accuracy, showing that preserving local temporal continuity benefits multimodal reasoning for both uniform sampling and key frames.

These results motivate our proposed method, which generalizes this idea by selecting key clips, thereby capturing both semantic relevance and temporal structure.

## 4 Proposed Method

Since temporal information can better help the QA task, we propose F2C, from key frame selection to a perspective of key clip selection. We decompose the key clips selection into two subtasks: 1) Select $K_{\text{anchor}}$ initial key frames that become the center of the key clips (*anchor key frames*), $\{C_1, C_2, \ldots, C_{K_{\text{anchor}}}\}$, and 2) Determine the clip length, $L_i$, for each anchor key frame to construct the key clip, as shown in Figure 3.

### 4.1 Anchor Key Frame Selection

To construct key clips, the first step is to determine the anchor key frames that serve as the centers of the clips. To control the diversity of the selected frames, we define the number of anchor key frames as $K_{\text{anchor}}$. We follow two guiding principles: (1) **Relevancy**, filtering out frames that are not related to the query; and (2) **Diversity**, ensuring that the selected anchors cover different parts of the video to avoid redundancy, especially since temporal neighbors will later be added to each anchor.

For relevancy, we leverage a pre-trained contrastive vision-language encoder $E$ (*e.g.*, CLIP (Radford et al., 2021) or SigLIP (Zhai et al., 2023; Tschannen et al., 2025)) to compute the cosine similarity between each frame $f_i$ and the text query $Q$ as

$$r(f_i) = \cos(E(f_i), E(Q)). \tag{2}$$

For multiple-choice questions, both the question and the candidate options are included in the query to avoid missing crucial clues.

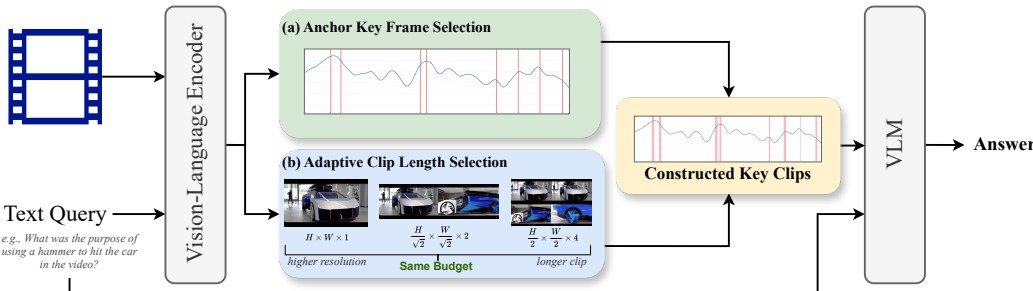

Figure 3: **Overview of F2C.** A video and text query are first processed by a vision-language encoder to select (a) *anchor key frames*. Each anchor is then extended into a short *key clip* through (b) adaptive clip length selection, which adjusts trade-off between resolution and clip length under the same budget. The constructed key clips preserve temporal continuity and, together with the query, are fed into a VLM to generate the answer.

Once relevancy scores are computed, anchor frames are selected by combining relevance with diversity. Inspired by the Watershed algorithm (Haralick et al., 1987), we first identify local maxima of the similarity curve (*i.e.*, watersheds) as an initial candidate pool. To further ensure diversity, we apply K-means clustering on the temporal indices of these candidates into $K_{\text{anchor}}$ clusters, and select within each cluster the frame with the highest similarity score as the anchor key frame. We provide a pseudo-code of the algorithm in Appendix D.2.

## 4.2 ADAPTIVE CLIP LENGTH SELECTION

### 4.2.1 EXPLORING TRADE-OFF BETWEEN RESOLUTION AND CLIP LENGTH

After selecting the anchor key frames, we analyze the trade-off between frame resolution and clip length under a fixed token budget. Consider a selection of $K$ frames of $f_i \in \mathbb{R}^{H \times W}$. In this case, our budget, $B$, can be formulated as

$$B = \frac{K \cdot H \cdot W}{Z}, \tag{3}$$

where $Z$ is the number of pixels per token. As seen in the equation, the total number of visual tokens is proportional to both $K$ and their spatial resolution, $H$ and $W$.

Now consider constructing $K_{\text{anchor}}$ anchor frames, each downsampled by a factor of $s$ to resolution $(H/s, W/s)$, and extending each anchor into a key clip of length $L$. The resulting token budget becomes

$$B_{\text{clip}} = \frac{K \cdot L \cdot (H/s) \cdot (W/s)}{Z} = \frac{K_{\text{anchor}} \cdot L}{s^2 \cdot K} \cdot B. \tag{4}$$

To keep the overall token budget unchanged (*i.e.*, $B_{\text{clip}} = B$), the clip length must satisfy

$$L = \frac{s^2 \cdot K}{K_{\text{anchor}}}. \tag{5}$$

This relation reveals a fundamental trade-off: longer clips can only be obtained by either reducing spatial resolution or decreasing the number of anchors, while higher-resolution clips limit the temporal context that can be included. This observation motivates our adaptive resolution strategy, which dynamically balances resolution and clip length under a fixed budget.

### 4.2.2 CLIP-SPECIFIC SELECTION

In the previous subsection, we applied a fixed clip length $L$ for all clips, which treats them equally and ignores their distinct characteristics. To address this, we extend $L$ into a clip-specific length $l_i$ for each clip $C_i$, centered at anchor frame $k_i$. Similar to Equation 5, $l_i$ is constrained by its scaling factor $s_i$ as $l_i = \frac{s_i^2 K}{K_{\text{anchor}}}$.

Table 2: Evaluation results of VQA accuracy across different frame budgets on Video-MME, LongVideo Bench, and MLVU. Missing results are shown as "-". † means our re-implementation results. Methods with ‡ uses various resolutions, and their budgets are equivalent to the reported # frames with the original resolution.

| Method | Size | # Frames | Video-MME (w.o./w. sub.) | | | | LongVideo Bench | MLVU |
| | | | Short | Medium | Long | Overall | | |
| *Avg. Video Duration* | | | *1.3min* | *9min* | *41min* | *17min* | *12min* | *12min* |
| *VLM w/ Uniform Sampling:* | | | | | | | | |
| Video-LLaVA (Lin et al., 2023a) | 7B | 8 | 45.3 / 46.1 | 38.0 / 40.7 | 36.2 / 38.1 | 39.9 / 41.6 | 39.1 | 47.3 |
| Qwen-VL (Bai et al., 2023) | 7B | 8 | 46.9 / 47.3 | 38.7 / 40.4 | 37.8 / 37.9 | 41.1 / 41.9 | - | - |
| VideoChat2 (Li et al., 2023) | 7B | 8 | 48.3 / 52.8 | 37.0 / 39.4 | 33.2 / 39.2 | 39.5 / 43.8 | 39.3 | 44.5 |
| Chat-UniVi-V1.5 (Jin et al., 2024) | 7B | 8 | 45.7 / 51.2 | 40.3 / 44.6 | 35.8 / 41.8 | 40.6 / 45.9 | - | - |
| VideoLLaMA2 (Cheng et al., 2024) | 7B | 16 | 56.0 / - | 45.4 / - | 42.1 / - | 47.9 / - | - | - |
| LLaVA-NeXT-QW2 (Liu et al., 2024a) | 7B | 8 | 58.0 / - | 47.0 / - | 43.4 / - | 49.5 / - | - | - |
| LongVILA (Xue et al., 2024) | 8B | 128 | 60.2 / - | 48.2 / - | 38.8 / - | 49.2 / - | - | - |
| LongVA (Zhang et al., 2024c) | 7B | 128 | 61.1 / 61.6 | 50.4 / 53.6 | 46.2 / 47.6 | 52.6 / 54.3 | - | - |
| Video-XL (Shu et al., 2024) | 7B | 128/256 | 64.0 / 67.4 | 53.2 / 60.7 | 49.2 / 54.9 | 55.5 / 61.0 | - | 64.9 |
| LLaVA-OneVision (Li et al., 2024) | 7B | * | 64.0 / 67.4 | 53.2 / 60.7 | 49.2 / 54.9 | 58.2 / - | 56.3 | 64.7 |
| Video-CCAM (Fei et al., 2024) | 9B | 96 | 61.9 / 63.1 | 49.2 / 52.3 | 39.6 / 42.4 | 50.3 / 52.6 | - | 58.5 |
| LongVU (Shen et al., 2024) | 7B | 1fps | 64.7 / - | 58.2 / - | 59.5 / - | 60.9 / - | - | 65.4 |
| *Training-based Selection:* | | | | | | | | |
| Frame-Voyager (Yu et al., 2024) | 7B | 8 | 67.3 / - | 56.3 / - | 48.9 / - | 57.5 / - | - | 65.6 |
| Hu et al. (2025) | 8.5B | 128 | 69.6 / - | 54.1 / - | 51.9 / - | 58.7 / - | - | - |
| GenS (Yao et al., 2025) | 7B | 54 | - / - | - / - | - / - | - / - | 58.7 | 64.8 |
| *Training-free Selection:* | | | | | | | | |
| **Qwen2.5-VL – 8 Frames** | | | | | | | | |
| *+ Uniform* | 7B | 8 | 60.9 / 68.6 | 51.4 / 56.2 | 47.4 / 51.2 | 53.3 / 58.7 | 53.3 | 52.8 |
| *+ Top-k* | 7B | 8 | 66.8 / 71.0 | 55.6 / 58.0 | 47.7 / 51.0 | 56.7 / 60.0 | 58.7 | 61.1 |
| *+ BOLT* † (Liu et al., 2025) | 7B | 8 | 66.0 / 71.7 | 54.6 / 57.1 | 50.4 / 52.0 | 57.0 / 60.3 | 55.6 | 59.0 |
| *+ Q-Frame* † (Zhang et al., 2025) | 7B | 8‡ | 71.9 / 74.0 | 58.2 / 60.6 | 50.7 / 54.0 | 60.3 / 62.9 | 56.6 | 58.2 |
| *+ Watershed* | 7B | 8 | 67.2 / 69.4 | 54.8 / 58.6 | 48.0 / 51.8 | 56.7 / 59.9 | 56.8 | 60.0 |
| *+ Ours* | 7B | 8‡ | **72.4 / 74.1** | **60.7 / 63.1** | **51.2 / 55.0** | **61.4 / 64.1** | **58.9** | **63.1** |
| **Qwen2.5-VL – 16 Frames** | | | | | | | | |
| *+ Uniform* | 7B | 16 | 67.3 / 71.2 | 55.0 / 58.8 | 48.9 / 51.9 | 57.1 / 60.6 | 56.7 | 57.1 |
| *+ Top-k* | 7B | 16 | 70.2 / 72.0 | 57.4 / 59.1 | 50.8 / 51.7 | 59.5 / 60.9 | 59.1 | 63.9 |
| *+ BOLT* † (Liu et al., 2025) | 7B | 16 | 71.7 / 72.3 | 57.7 / 60.8 | 49.7 / 53.1 | 59.7 / 62.1 | 58.0 | 64.5 |
| *+ Q-Frame* † (Zhang et al., 2025) | 7B | 16‡ | 73.4 / 74.9 | 61.4 / 63.3 | 52.3 / 53.7 | 62.4 / 64.1 | 57.1 | 61.9 |
| *+ Watershed* | 7B | 16 | 70.0 / 72.1 | 58.3 / 62.7 | 50.7 / 52.4 | 59.7 / 62.4 | 56.8 | 62.3 |
| *+ Ours* | 7B | 16‡ | **73.8 / 75.6** | **62.2 / 65.6** | **54.1 / 56.1** | **63.4 / 65.7** | **61.1** | **65.5** |
| **Qwen2.5-VL – 32 Frames** | | | | | | | | |
| *+ Uniform* | 7B | 32 | 72.6 / 73.7 | 59.0 / 62.6 | 51.8 / 55.1 | 61.1 / 63.8 | 58.4 | 59.4 |
| *+ Top-k* | 7B | 32 | 74.2 / **76.4** | 59.9 / 62.0 | 51.9 / 54.0 | 62.0 / 64.1 | 60.1 | 66.6 |
| *+ BOLT* † (Liu et al., 2025) | 7B | 32 | **74.3** / 76.2 | 64.2 / 63.9 | 53.8 / 56.4 | 64.1 / 65.5 | 58.6 | 66.3 |
| *+ Q-Frame* † (Zhang et al., 2025) | 7B | 32‡ | 73.0 / 75.0 | 61.7 / 62.9 | 53.1 / 53.7 | 62.6 / 63.9 | 58.7 | 41.6 |
| *+ Watershed* | 7B | 32 | 71.8 / 73.6 | 60.2 / 63.4 | 52.1 / 55.2 | 61.4 / 64.1 | 58.5 | 64.0 |
| *+ Ours* | 7B | 32‡ | 73.8 / 75.9 | **66.3 / 68.1** | **56.6 / 57.8** | **65.6 / 67.3** | **60.8** | **66.8** |

To reduce the search space, we introduce a maximum scaling factor $s_{max}$ as a hyperparameter, which defines the largest possible clip length $l_{max} = \frac{s_{max}^2 K}{K_{anchor}}$. Each clip is then optimized independently within the range $[1, l_{max}]$.

For a candidate clip length $l$, the clip $C_i$ spans the interval $[k_i - \lfloor (l-1)/2 \rfloor, k_i + \lfloor (l-1)/2 \rfloor]$. Its importance is measured by three components:

**Relevancy.** We compute the average cosine similarity between the frames in $C_i$ and the text query:

$$S_C(l) = \frac{1}{l} \sum_i r(f_i), \tag{6}$$

where $r(f_i)$ is defined in Equation 2.

**Redundancy.** To discourage redundant frames, we compute the average pairwise similarity between frames:

$$R_C(l) = \frac{1}{l(l-1)} \sum_i \sum_{j \neq i} \cos\big(E(f_i), E(f_j)\big), \tag{7}$$

where $E$ is the vision-language encoder.

**Temporal reward.** We encourage longer clips to compensate for potential noise in similarity scores by adding a reward proportional to the relative clip length $l/l_{max}$.

**Optimization objective.** Combining the above, the clip is determined by finding $l^*$ that optimizes the following by exhaustive search

$$l^* = \arg\max_{1 \leq l \leq l_{max}} \Big(S_C(l) - \lambda_r R_C(l) + \lambda_l \frac{l}{l_{max}}\Big), \tag{8}$$

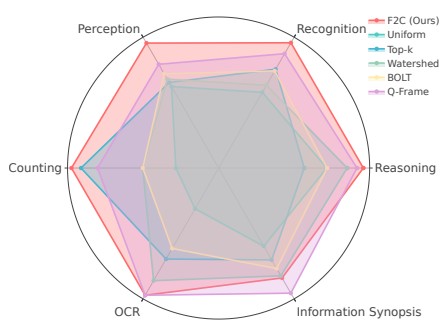

Figure 4: Performance on each type of questions on Video-MME with 16 frames.

Table 3: Ablation study on Video-MME.

| Method | Short | Medium | Long | Overall |
|---|---|---|---|---|
| Uniform | 67.3 | 55.0 | 48.9 | 57.1 |
| Watershed | 70.0 | 58.3 | 50.7 | 59.7 |
| + Key Clip (Fixed) | 73.0 | 60.0 | 53.7 | 62.2 |
| + Key Clip (Adaptive) | **73.8** | **62.2** | **54.1** | **63.4** |

Table 4: Comparison of temporal modeling on Video-MME. Baselines use 64 isolated frames, while F2C uses 16 clips of 4 frames.

| Method | # Frames | Short | Medium | Long | Overall |
|---|---|---|---|---|---|
| Uniform | 64 | 71.3 | 60.1 | 51.4 | 61.0 |
| Top-$k$ | 64 | 73.6 | 59.9 | 52.3 | 61.9 |
| BOLT | 64 | 73.4 | 61.0 | 50.6 | 61.7 |
| Watershed | 64 | 72.3 | 59.8 | 53.4 | 61.9 |
| **F2C (Ours)** | 16×4 | **73.8** | **62.2** | **54.1** | **63.4** |

where $\lambda_r$ and $\lambda_l$ control the scales between each score. The corresponding scaling factor is then computed as $s_i^* = \sqrt{\frac{K_{\text{anchor}} \cdot l_i^*}{K}}$ to obtain the spatial resolution of the current clip.

Finally, overlapping clips with identical resolutions are merged to avoid redundancy, yielding the set of key clips that are passed into the VLM.

## 5 EXPERIMENTS

### 5.1 EXPERIMENTAL SETTINGS

**Benchmarks.** We evaluate our method on three widely used long-form video benchmarks: Video-MME, LongVideoBench, and MLVU, which together cover diverse domains and evaluation protocols. *Video-MME* (Fu et al., 2024) is a large-scale multimodal evaluation suite designed to assess the capabilities of Video LLMs across video understanding, reasoning, and knowledge-grounded tasks. It includes both short and long-form videos, enabling a comprehensive evaluation of context management strategies. *LongVideoBench* (Wu et al., 2024b) specifically targets long-form video understanding with a diverse set of tasks such as VQA, summarization, and temporal grounding. It emphasizes scenarios where videos span tens of minutes to hours, providing a challenging testbed for evaluating temporal reasoning and efficient context construction. *MLVU* (Multi-level Long Video Understanding) (Zhou et al., 2024) is a benchmark focused on hierarchical understanding of long videos. It evaluates models on multiple levels of reasoning, including frame-level recognition, clip-level temporal understanding, and video-level holistic comprehension. This makes MLVU particularly suitable for testing whether methods like ours preserve both local and global context.

**Comparison Methods.** We compare our proposed method with several representative selection strategies which are applicable to long-form videos. *Uniform sampling* selects frames at equal temporal intervals across the video. *Top-k sampling* selects the frames with the highest similarity scores to the given text query. The training-free method BOLT (Liu et al., 2025) selects frames using Inverse Transform Sampling based on frame importance scores. Q-frame (Zhang et al., 2025) selects three levels of frames in different resolutions. In addition, we include watershed selection, which selects frames according to temporal boundaries detected by watershed segmentation.

**Implementation Details.** We use Qwen2.5-VL-7B as the backbone VLM due to its support for any input resolution. All videos are loaded at 1 FPS for selection. For all baseline methods, we keep the original video resolution without resizing to ensure a fair comparison. Evaluations are conducted using the lmm_evals library (Zhang et al., 2024b) on a computing cluster equipped with NVIDIA A100 GPUs. For similarity-based selection (Top-$k$), we use SigLIP2 (Tschannen et al., 2025) as the vision-language model to compute similarity scores. More details are provided in Appendix D.

### 5.2 RESULTS ON LONG-FORM VIDEO BENCHMARKS

Table 2 summarizes the results on Video-MME, LongVideoBench, and MLVU under different frame budgets. We compare F2C with uniform sampling, Top-$k$, watershed, Q-Frame, and BOLT.



Table 5: Token number analysis on Video-MME under different budgets of full-resolution frames.

| Method | 8 Frames | 16 Frames | 32 Frames |
|---|---|---|---|
| Baselines | 4428.97 | 8276.95 | 17259.13 |
| Ours (F2C) | 4359.95 (-1.6%) | 8269.90 (-0.1%) | 14704.81 (-14.8%) |

Figure 5: Impact of different initial keyframe selectors on Video-MME with $K = 16$.

Figure 6: Performance across different $K$ and $K_{\text{anchor}}/K$ ratios on Video-MME.

Across all the three benchmarks, F2C consistently surpasses baselines. Compared with uniform sampling, it yields substantial improvements, with the largest gains on MLVU, followed by Video-MME, and smaller but steady gains on LongVideoBench. The benefit is most pronounced under small frame budgets (e.g., $K = 8$), where uniform sampling often misses critical content, while F2C remains robust by preserving temporal continuity. As $K$ increases, the relative gap narrows, but F2C still retains a stable advantage.

Compared with Q-Frame, which primarily adjusts resolution at the frame level, F2C extends selection to temporally coherent clips. This integration of adaptive resolution and temporal continuity provides richer context and consistently stronger results. Overall, these findings demonstrate that F2C offers a training-free and scalable solution for long-form video understanding.

### 5.3 RESULTS ON DIFFERENT TYPES OF QUESTIONS

To better understand F2C, we break down the results of Video-MME ($K = 16$) into six question categories, as shown in Figure 4, where we scale the range on each axis to $0.9\times$ min to $1.05\times$ max. Compared to uniform sampling, Top-$k$, watershed, BOLT, and Q-Frame, F2C achieves consistent improvements across all categories. The gains are especially large in *Counting*, *Recognition*, and *Reasoning*, where temporal continuity and adaptive resolution provide richer motion and contextual cues. F2C also excels in *Perception* and *OCR*, indicating its ability to balance fine-grained detail with temporal coherence. Overall, these results demonstrate the benefits of F2C extend broadly across diverse question types, confirming its general effectiveness for long-form video understanding.

## 6 DISCUSSION

### 6.1 ABLATION STUDY

We conduct an ablation study on Video-MME with $K = 16$ (Table 3), comparing uniform sampling, watershed selection, fixed-length key clips, and adaptive key clips. Watershed and key clip selection both outperform uniform sampling, showing the benefit of more informed context construction. Key clips are especially effective on long videos by preserving local temporal continuity, and adaptive resolution yields the best overall performance by balancing temporal coverage with spatial detail.

### 6.2 IMPACT OF TEMPORAL INFORMATION

To examine whether temporal continuity provides greater benefit than simply sampling more frames, we conduct an experiment with a total of 64 frames. For baselines, 64 isolated frames are selected directly, while in F2C the same budget is allocated to 16 clips with up to 4 frames each. To ensure fairness, all frames are downsampled with $s = 2$, so the token count remains comparable across methods. Results are reported in Table 4.

F2C consistently outperforms the baselines, with the largest gain on long videos, where it reaches 54.1 compared to 50.6–53.4 for separate-frame approaches. This demonstrates that grouping frames into clips yields richer temporal cues than simply increasing frame count. Moreover, since F2C

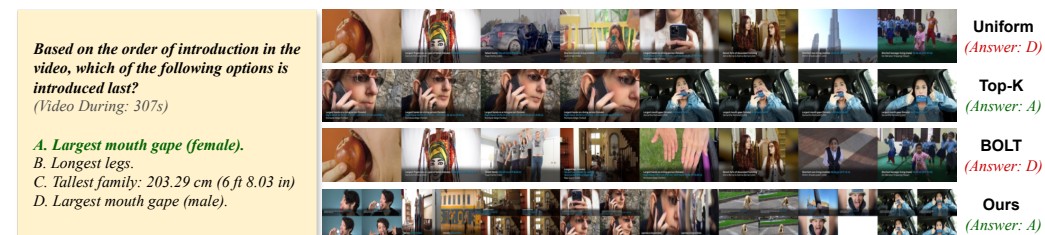

Figure 7: Visualization of the selected frames for each method.

introduces resolution as an additional factor in context construction, the results also indicate that resolution-aware design can further enhance existing frame selection methods.

### 6.3 DISCUSSION ON ANCHOR KEY FRAME SELECTION

**Selectors.** In F2C, we default to watershed for anchor keyframe selection, but other strategies can also be used. We apply F2C on top of uniform sampling, Top-$k$, BOLT, and watershed. As shown in Figure 5, F2C consistently improves performance across all selectors, confirming the general benefit of extending anchors into clips. However, the quality of anchors matters: Top-$k$, which yields low-diversity frames, performs worst even after enhancement, while BOLT and watershed provide stronger results due to higher diversity. Uniform sampling benefits from diversity but lacks semantic relevance, limiting its effectiveness. These findings highlight that both diversity and relevance are crucial for anchor detection, and F2C effectively leverages them.

**Impact of $K_{\text{anchor}}$.** We further study the number of anchor frames by varying $K_{\text{anchor}}/K$, where $K$ is the total frame budget. Results in Figure 6 show that smaller $K_{\text{anchor}}$ reduces diversity, while larger $K_{\text{anchor}}$ sacrifices temporal continuity. The best balance is achieved when $K_{\text{anchor}} = K$, which preserves both spatial detail and temporal coverage.

### 6.4 COMPUTATIONAL ANALYSIS

We analyze the computational efficiency of F2C by comparing visual token counts on Video-MME (w/o subtitles) under budgets of 8, 16, and 32 full-resolution frames (Table 5). Despite adding temporal context, F2C achieves similar or fewer tokens than frame-based selectors because overlapping frames are encoded only once. The reduction becomes increasingly significant as the budget grows, showing that F2C not only enriches temporal continuity but also improves efficiency. This balance of accuracy and computation makes F2C well suited for long-form video understanding.

### 6.5 VISUALIZATION

Figure 7 provides a qualitative comparison of different frame selection strategies on a long-form video. Uniform sampling and BOLT fail to capture the crucial frames corresponding to the correct answer, resulting in incorrect predictions. Top-K manages to select one relevant frame but lacks sufficient coverage of the question, leading to incomplete context. In contrast, our method successfully selects the key frames related to the correct option while also maintaining high diversity across the video, enabling more comprehensive temporal reasoning and accurate prediction. Besides, we provide a visualization on the distribution of the selections in Appendix E.

### 7 CONCLUSION

We revisited the frame selection for long-form video understanding and proposed Frames-to-Clips, a training-free framework that replaces isolated keyframes with temporally coherent clips and introduces adaptive trade-off between resolution and clip length under a fixed token budget. Extensive experiments on Video-MME, LongVideoBench, and MLVU show that F2C consistently outperforms state-of-the-art training-free baselines. Our analyses further demonstrate that temporal continuity, diversity in anchor selection, and balanced resolution scaling are all crucial for effective context management. By improving selection without training, F2C offers a simple and scalable solution for long-form video understanding and points toward future directions for VLMs.

ETHICS STATEMENT

This work adheres to the ICLR Code of Ethics. Our study relies only on publicly available video benchmarks (Video-MME, LongVideoBench, and MLVU), which do not involve personally identifiable information or sensitive human subject data. We do not release or collect new data, and our method is purely training-free, avoiding additional computational cost or energy overhead beyond standard inference. Potential applications include video understanding tasks such as question answering and reasoning. We acknowledge that, as with all video-language technologies, misuse may raise privacy or surveillance concerns; however, our contributions are limited to context management strategies and do not enable new forms of video capture or analysis.

REPRODUCIBILITY STATEMENT

We have taken several steps to ensure reproducibility. Implementation details, including model configuration, datasets, evaluation protocols, and hyperparameter settings, are described in Section 5.1 and Appendix D. Additional studies and analyses are provided in Section 6. All datasets used in this work are publicly available and properly cited.

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

APPENDIX

## A  OVERVIEW

This appendix provides additional details and supporting analyses to complement the main paper. Specifically, we include:

- **Usage of Large Language Models** (Section B): clarifies how LLMs were used to assist in writing and formatting.
- **Limitations** (Section C): discusses constraints of F2C and dependency on backbone VLMs.
- **Implementation Details** (Section D): provides prompts, pseudo-code for watershed selection, and re-implementation details of BOLT and Q-Frame.
- **Visualization of Selection Distributions** (Section E): illustrates qualitative comparisons of frame selection strategies.
- **Additional Experiments** (Section F.1): analyzes the impact of different CLIP backbones on performance.

## B  USAGE OF LARGE LANGUAGE MODEL

We used a Large Language Model (LLM) to assist in polishing the writing style and generating LaTeX code for tables. All technical ideas, experimental designs, and analyses were developed by the authors.

## C  LIMITATION

Although F2C effectively improves context management by providing temporally coherent and resolution-adaptive inputs, its performance upper bound is still constrained by the capability of the downstream VLM. Even when the correct frames or clips are selected, the final answer depends on the reasoning and comprehension ability of the VLM itself. As a result, limitations in temporal reasoning, spatial understanding, or multimodal alignment within the backbone model remain bottlenecks. Our method is thus complementary to future advances in Video LLM architectures, and its benefits may further amplify as more powerful backbone models become available.

## D  IMPLEMENTATION DETAILS

### D.1  PROMPT FOR VQA

We use the following prompt to query VLM for the VQA task with temperature of 0.0 for reproducibility.

---
**Prompt for Video Question Answering**

<Selected Frames>

Select the best answer to the following multiple-choice question based on the video and the subtitles. Respond with only the letter (A, B, C, or D) of the correct option.

<Questions>

<Options>

Answer with the option's letter from the given choices directly.

---

We use the default random seeds from `lmms_eval` for `random`, `numpy`, and `torch` as $0, 1234$, and $1234$, respectively.

---

**Algorithm 1:** Watershed Selection

---

**Input:** Frame features $\{f_i\}_{i=1}^N$, text feature $q$, number of anchors $K_{\text{anchor}}$
**Output:** Indices of selected anchor frames
- Normalize frame and text features, compute similarities
$\quad s_i = \cos(f_i, q)$.
- Find valleys (local minima) in the similarity curve to define basin boundaries.
- For each basin, identify the peak frame (highest similarity) as a candidate.
- **If** number of candidates $> K_{\text{anchor}}$:
$\quad$ Cluster candidates into $K_{\text{anchor}}$ groups (via k-means).
$\quad$ In each cluster, select the frame with the highest similarity.
- **Else:**
$\quad$ Use all candidates (up to $K_{\text{anchor}}$).
**Return** the sorted list of selected anchor frames.

---

### D.2 PSEUDO-CODE OF WATERSHED SELECTION

We provide the pseudo-code of our watershed-based anchor keyframe selection strategy in Algorithm 1. This procedure identifies local maxima of frame–text similarity within basins, then clusters them if necessary to ensure both diversity and relevance of the selected anchors.

### D.3 HYPERPARAMETER OF F2C

We choose $\lambda_r = 0.5$ and $\lambda_l = 0.05$ for the adaptive clip length selection and set $s_{\max} = 2$ and $K_{\text{anchor}} = K$ for the experiments.

### D.4 RE-IMPLEMENTATION OF BOLT AND Q-FRAME

For a fair comparison, we closely followed the official settings reported in the original papers. For BOLT (Liu et al., 2025), we adopted the same hyperparameters as ours and set $\alpha = 2.5$ as recommended in their work. For Q-Frame (Zhang et al., 2025), the original implementation evaluates three configurations of $(\text{num}_{\text{high}}, \text{num}_{\text{medium}}, \text{num}_{\text{low}}) = (4, 8, 32)$, which corresponds to our 8-frame setting. To extend Q-Frame under larger budgets, we applied proportional scaling and used $(8, 16, 64)$ for the 16-frame setting and $(16, 32, 128)$ for the 32-frame setting. These adjustments ensure that both baselines are re-implemented consistently and evaluated within our experimental framework, facilitating fair comparison and reproducibility.

## E VISUALIZATION ON DISTRIBUTION OF SELECTION

To complement the qualitative examples in the main paper, we provide an additional visualization of how different methods distribute their selected frames across the video (Figure 8). The similarity curve represents the frame-level relevance to the query.

Uniform sampling spreads frames evenly but ignores semantic signals, often missing important peaks. BOLT tends to favor high-similarity regions but suffers from redundancy. Top-$k$ focuses too narrowly, selecting clustered frames around peaks while overlooking other segments. In contrast, F2C selects temporally coherent clips that cover diverse regions with high relevance, striking a better balance between coverage and precision.

This visualization further illustrates that F2C not only captures key peaks but also preserves temporal continuity, leading to more informative and efficient context construction.

## F MORE EXPERIMENTS

### F.1 IMPACT OF THE VERSION OF CLIP MODEL

We observe that the choice of CLIP backbone has only a marginal impact on performance. As shown in Table 6, all versions achieve similar accuracies across different frame budgets (8, 16,

Table 6: Comparison of different CLIP versions on Video-MME overall accuracy (%). Columns show performance under 8, 16, and 32 frames.

| CLIP-version | Max length | 8 | 16 | 32 |
|---|---|---|---|---|
| LongCLIP (Zhang et al., 2024a) | 248 | 61.9 | 63.1 | 65.1 |
| CLIP (Radford et al., 2021) | 77 | 61.9 | 65.1 | 65.7 |
| SigLIP (Zhai et al., 2023) | 64 | 62.5 | 64.8 | 65.4 |
| SigLIP2 (Tschannen et al., 2025) | 64 | 61.4 | 63.4 | 65.6 |

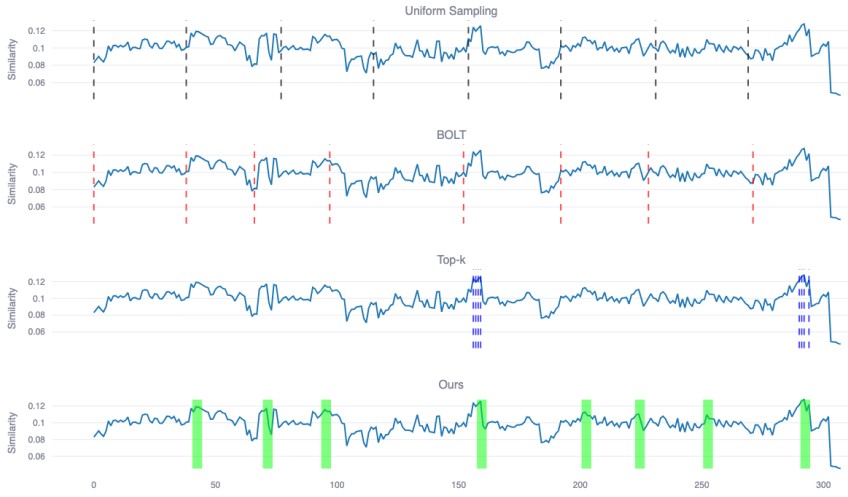

Figure 8: Visualization of the selected frames

and 32). While CLIP slightly outperforms others at 16 and 32 frames, and SigLIP leads under the 8-frame setting, the overall differences remain within a narrow margin (typically less than 2%). This suggests that the improvements brought by our method are robust to the underlying vision-language encoder choice, and the effect of CLIP version is not a dominant factor in determining downstream performance. In our experiments, we adopt SigLIP2 as the default backbone since it offers competitive accuracy while being more efficient at shorter sequence lengths, and it represents a more recent CLIP variant.

