# OpenReview forum: "From Frames to Clips: Efficient Key Clip Selection for Long-Form Video Understanding"
_ICLR.cc/2026/Conference — ICLR 2026 Conference Withdrawn Submission_

### Official Review · Reviewer_oqSe · 2025-10-28

**Soundness:** 2
**Presentation:** 2
**Contribution:** 2
**Rating:** 6
**Confidence:** 3

**Summary:**

Several methods utilize keyframe selection for long video understanding to reduce computational burden by processing only the most relevant frames. However, these mechanisms are neglecting the surrounding temporal context. This work proposes leveraging key clip selection instead of single frames, arguing that crucial temporal information may reside in adjacent frames. The authors also investigate the trade-off between decreasing video spatial resolution to increase clip length versus maintaining high spatial resolution with shorter clip lengths. By selecting sub-clips, using either fixed or variable lengths, they demonstrate significant improvement across various benchmarks, including Video-MME, LongVideoBench, and MLVU.

**Strengths:**

- The motivation is sounded. Selecting subclip instead of individual frames make indeed more sense for video understanding.
- Really appreciate the tradeoff analysis between the spatial and temporal resolution. It's true that in some case it might be more effective to decrease the spatial resolution while increasing the temporal one.
- Good ablation study over using either a fixed or adapative strategy for clip length, and ablation over the vision encoder.
- Training free method

**Weaknesses:**

- Only use Qwen2.5-VL as base VLM, not sure how the method will work on other VLMs. Are other VLMs such as Llava-Video not supporting variable spatial resolution?
- Reliance on other model such as SigLip2 to compute similarity score add a computational burden. Not sure if it was considered in 6.4 on Computational analysis.
- I am surprised to not see AKS in Table2.

**Questions:**

Did you run any experiment with other VLMs than Qwen2.5-VL?

---

### Official Review · Reviewer_D9X5 · 2025-10-29

**Soundness:** 2
**Presentation:** 2
**Contribution:** 2
**Rating:** 4
**Confidence:** 4

**Summary:**

This paper proposes a training-free framework that replaces isolated key frames with temporally coherent key clips to better capture motion and temporal context under a fixed token budget.
By introducing an adaptive trade-off between clip length and spatial resolution, F2C consistently improves Video-LMM performance on multiple long-form video benchmarks, demonstrating superior efficiency and temporal reasoning ability.

**Strengths:**

1.	The paper introduces F2C that replaces isolated key frames with temporally coherent key clips, addressing the “needle in a haystack” problem in long-form video understanding.
2.	The method is technically sound and well-analyzed, offering an adaptive trade-off between spatial resolution and clip length, with clear ablations verifying the impact of each design choice.
3.	F2C shows consistent improvements across multiple benchmarks while remaining computationally efficient and easy to integrate into existing Video-LMM pipelines.

**Weaknesses:**

1. As introduced in the abstract, long-form video understanding tasks are constrained by the “needle in a haystack” problem. The authors should provide a clearer visualization or comparison to illustrate how their method alleviates this issue (e.g., similar to Figure 4 in [1], Figure 8 in [2], or visualization examples in [3]).

2. Compared with frame-level token merging methods [4], it remains unclear what specific advantages this token selection strategy provides.

3. The paper lacks some necessary references to related work, especially other training-free long-form video understanding models [5] [6] [7].

4. It is also unclear whether the λr and λl choices are dependent on the benchmark used.

[1] Zhang, Peiyuan, et al. "Long context transfer from language to vision." arXiv preprint arXiv:2406.16852 (2024).

[2] Zhao, Zijia, et al. "Needle in a video haystack: A scalable synthetic evaluator for video mllms." arXiv preprint arXiv:2406.09367 (2024).

[3] https://github.com/bigai-nlco/NeedleInAVideoHaystack

[4] Song, Enxin, et al. "Moviechat+: Question-aware sparse memory for long video question answering." IEEE Transactions on Pattern Analysis and Machine Intelligence (2025).

[5] Santos, Saul, et al. "$\infty $-Video: A Training-Free Approach to Long Video Understanding via Continuous-Time Memory Consolidation." arXiv preprint arXiv:2501.19098 (2025).

[6] Xu, Mingze, et al. "Slowfast-llava: A strong training-free baseline for video large language models." arXiv preprint arXiv:2407.15841 (2024).

[7] Zhang, Yiming, et al. "Beyond training: Dynamic token merging for zero-shot video understanding." Proceedings of the IEEE/CVF International Conference on Computer Vision. 2025.

**Questions:**

See weakness

---

### Official Review · Reviewer_HQSx · 2025-10-29

**Soundness:** 2
**Presentation:** 3
**Contribution:** 2
**Rating:** 2
**Confidence:** 4

**Summary:**

The paper proposes a frame-to-clips (F2C) framework that addresses frame selection for long-video understanding by first selecting anchor key frames and then determining corresponding clip lengths. The framework maintains a fixed computational budget through adaptive resolution adjustment. The authors also show that temporally coherent clips perform better for long-video understanding compared to isolated frames. The effectiveness of the F2C framework is demonstrated on several long-video understanding benchmarks, along with ablations of different frame-selection strategies.

**Strengths:**

The paper is well-organized and easy to read. It proposes a straightforward idea for frame selection in long-video understanding and shows the effectiveness of the approach. The method is training-free, making it practical for application to existing models for long-video VQA tasks. The paper also provides analysis of different key-frame and clip-selection strategies.

**Weaknesses:**

1. The novelty of the paper appears limited relative to the practical engineering effort. The method primarily relies on SigLIP2 embeddings computed for each frame for both key-frame and clip selection. It is unclear what the computational cost is for performing per-frame similarity comparisons at scale. Additionally, relying solely on SigLIP2 embeddings may fail to capture or aggregate sufficient temporal information, particularly for tasks requiring temporal reasoning. Although reducing the resolution helps maintain the token budget, this approach may result in the loss of important spatial details.
2. There are few strong baselines are missing for both training [1][2][3]/training-free [4]

[1] Yang, B., Wen, B., Ding, B., Liu, C., Chu, C., Song, C., Rao, C., Yi, C., Li, D., Zang, D. and Yang, F., 2025. Kwai keye-vl 1.5 technical report. arXiv preprint arXiv:2509.01563.

[2] Cheng, C., Guan, J., Wu, W. and Yan, R., 2025. Scaling Video-Language Models to 10K Frames via Hierarchical Differential Distillation. arXiv preprint arXiv:2504.02438.

[3] Xu, M., Gao, M., Li, S., Lu, J., Gan, Z., Lai, Z., Cao, M., Kang, K., Yang, Y. and Dehghan, A., 2025. Slowfast-llava-1.5: A family of token-efficient video large language models for long-form video understanding. arXiv preprint arXiv:2503.18943.

[4] Xu, M., Gao, M., Gan, Z., Chen, H.Y., Lai, Z., Gang, H., Kang, K. and Dehghan, A., 2024. Slowfast-llava: A strong training-free baseline for video large language models. arXiv preprint arXiv:2407.15841.

**Questions:**

1. Could the authors provide experiments or analysis on the latency of performing per-frame similarity computation across long videos? It would be particularly helpful to understand scalability when videos contain thousands of frames.
2. I have concerns about whether temporal information can be fully captured when relying solely on SigLIP2 embeddings. EgoSchema is a challenging benchmark that includes complex temporal reasoning. Could the authors comment on how well their approach would generalize to such benchmarks?

---

### Official Review · Reviewer_2WqS · 2025-10-29

**Soundness:** 3
**Presentation:** 2
**Contribution:** 1
**Rating:** 4
**Confidence:** 5

**Summary:**

The paper proposes a key clip selection framework, F2C, that integrates adjacent frames to restore temporal continuity, together with an adaptive resolution scheme to mitigate token overhead. It conducts a comprehensive analysis of frame selection strategies and show that selecting temporally coherent clips, rather than isolated frames, significantly improves VLM performance on long-form videos.

**Strengths:**

+ The proposed method is training-free and efficient.
+ The proposed formulation to balance spatial resolution and clip length under a fixed token budget.
+ Evaluated on multiple long-video benchmarks (Video-MME, LongVideoBench, MLVU) with consistent improvements.

**Weaknesses:**

+ Some notations are not clear.
+ Missing some training-free key frame selection baselines, for example AKS [1] and CoS [2].
+ Limited novelty and contribution. While the shift from “frames” to “clips” is effective, it is conceptually incremental, relying mostly on a combination of known techniques (keyframe selection + adaptive resolution).

[1] Adaptive Keyframe Sampling for Long Video Understanding, CVPR 2025.
[2] CoS: Chain-of-Shot Prompting for Long Video Understanding, arxiv:2502.06428.

**Questions:**

+ Equation 4 is confusing. Shouldn't it be $B_{clip}=\frac{K_{anchor}\cdot L \cdot (H/s) \cdot (W/s)}{Z}$ instead of $B_{clip}=\frac{K \cdot L \cdot (H/s) \cdot (W/s)}{Z}$?
+ How about the performance by extending Top-k frames to key clips, e.g., 64 key frames, each key frame extended with 4 frames.
+ How many frames per clip are used on average for each benchmark? As the method shows, the clip length varies based on relevancy and redundancy. But in Table 4, it seems that F2C uses a fixed number of frames as 4.
+ In line 779, it shows $K_{anchor} = K$. Isn't the $K_{anchor} << K$ since as line 237 mentioned, $K_{anchor}$ are the clusters from the selected candidate frames.

---

### Note · Authors · 2025-11-14

I have read and agree with the venue's withdrawal policy on behalf of myself and my co-authors.